# Towards Eliminating Hard Label Constraints in Gradient Inversion Attacks

**Yanbo Wang**[1,2]**, Jian Liang**[1,2]**, Ran He**[1,2*]
[1]School of Artificial Intelligence, University of Chinese Academy of Sciences (UCAS)
[2]CRIPAC & MAIS, Institute of Automation, Chinese Academy of Sciences (CASIA)
`wangyanbo2023@ia.ac.cn, liangjian92@gmail.com, rhe@nlpr.ia.ac.cn`

## Abstract

Gradient inversion attacks aim to reconstruct local training data from intermediate gradients exposed in the federated learning framework. Despite successful attacks, all previous methods, starting from reconstructing a single data point and then relaxing the single-image limit to batch level, are only tested under hard label constraints. Even for single-image reconstruction, we still lack an analysis-based algorithm to recover augmented soft labels. In this work, we change the focus from enlarging batchsize to investigating the hard label constraints, considering a more realistic circumstance where label smoothing and mixup techniques are used in the training process. In particular, we are the first to initiate a novel algorithm to simultaneously recover the ground-truth augmented label and the input feature of the last fully-connected layer from single-input gradients, and provide a necessary condition for any analytical-based label recovery methods. Extensive experiments testify to the label recovery accuracy, as well as the benefits to the following image reconstruction. We believe soft labels in classification tasks are worth further attention in gradient inversion attacks[1].

## 1 Introduction

Federated or distributed learning (Shi et al., 2022; Luo et al., 2021) enables multiple participants to train one specific model collaboratively so as to increase efficiency and settle privacy concerns (McMahan et al., 2017; Yang et al., 2019; Li et al., 2014). During the information sharing process, only intermediate gradients or parameters are transmitted to the central server under a common horizontal federated learning paradigm, which is believed to effectively protect the input data privacy (Lyu et al., 2022; Wei et al., 2020; Zhang et al., 2022).

Even if federated learning is developed for its privacy protection capability, recent works have proved that with accurate gradient information, input training data could be reconstructed under ideal circumstances (Zhu & Blaschko, 2020; Wang et al., 2020) through gradient inversion attacks (GIA). Representing works achieve such a goal mainly by establishing a gradient matching process, using the matching loss between dummy gradients from randomly initialized input data and ground-truth gradients to optimize dummy data towards ground-truth local data (Geiping et al., 2020; Yin et al., 2021; Zhu et al., 2019).

When revisiting the series of works, we find that their contributions mainly focus on enlarging the batchsize. They try various gradient matching functions, different regularization terms (Geiping et al., 2020; Yin et al., 2021), and label recovery algorithms (Zhao et al., 2020; Ma et al., 2023) to increase the reconstruction accuracy. Therefore, works after DLG (Zhu et al., 2019) split the matching process into two steps: recover the one-hot label at first and then optimize the input image based on ground-truth labels. **However, has the single-image reconstruction task been perfectly solved?** The answer is negative, and we still identify crucial limitations for the basic single-image GIA when faced with soft labels. In real-world settings, label augmentation techniques, such as label smoothing (Szegedy et al., 2016) and mixup (Zhang et al., 2018a), are generally used to enhance

---

[*]Corresponding author

[1]Our code is publicly available at `https://github.com/ybwang119/label_recovery`.

model robustness and generalization capability. Previous works, relying on the sign of gradients to decide the ground-truth one-hot label (Zhao et al., 2020; Yin et al., 2021), are unable to handle augmented labels, for they will disable the sign indicator. Consequently, these label augmentations may serve as a simple defense against GIA (Huang et al., 2021), even though they are not designed to enhance model safety. Besides, previous literature also mentioned that for fully-connected networks (FCN), input data could only be reconstructed from gradients analytically when layers have non-zero bias term (Aono et al., 2017). Zhu & Blaschko (2020) propose a recursive analytical data recovery algorithm regardless of the bias term, but it only works for binary classification tasks. Without strict limitations as above, current analytical methods cannot even handle the simplest FCN.

Aware of such constraints in single-image label recovery, this work initiates a novel algorithm to retrieve accurate augmented labels as well as the last layer features from corresponding gradients, regardless of the existence of the bias term. We identify mathematical features of general multi-class classification tasks with cross-entropy loss and successfully break the label reconstruction task into solving one scalar from equations. To figure out the specific scalar, a simple but effective label reconstruction loss is adopted to search for it while trying to avoid local minima. Once such a scalar is solved, both the input feature and the augmented label can be recovered precisely. Extensive experiments on various datasets under multiple networks demonstrate the correctness of such a scalar.

Based on our algorithm, we then explain why the bias term matters for previous feature recovery methods and point out a necessary condition to analytically recover augmented labels. Furthermore, convincing comparative experiments are designed to evaluate the image reconstruction profits from precisely recovered augmented labels. We conclude that the proposed method could raise image reconstruction accuracy to the level that is achieved with full knowledge of input labels.

To summarize, our contributions include:

- Initiating a novel analytical algorithm to accurately reconstruct the augmented label of the single input image with input features, and more importantly, disclosing a necessary condition for all analytical-based label recovery methods;

- Analyzing limitations for previous bias-based recovery methods, proposing the first method to analytically reconstruct input image from FCNs under multi-class classification tasks based on recovered last-layer features regardless of the bias term;

- Designing extensive experiments to demonstrate the label recovery accuracy and image reconstruction profits from our proposed algorithm, claiming that gradient inversion attacks could still be effective under real-world settings.

## 2 RELATED WORK

**Label recovery methods.** Previous methods mainly focus on retrieving one-hot labels from single images to batches. Zhao et al. (2020) propose iDLG, identifying that for multi-class classification with cross-entropy loss, the sign of last fully-connected layer gradients for the entry of ground-truth class is different from any other entries[2]. Yin et al. (2021) extend this theory to derive batched labels from the minimum value of each entry's gradients; Dang et al. (2021) propose RLG based on singular value decomposition to recover one-hot labels of a whole batch, which achieves better results; to nullify the limit that each class could at most have one image in a batch, Geng et al. (2021) improve iDLG, replacing unknown variables by the mean value to calculate labels for every class; Ma et al. (2023) then identify consistency in gradients of each class, constructing linear equations to solve instance-wise one-hot labels. However, for non-one-hot labels, such as label smoothing (Szegedy et al., 2016) and mixup (Zhang et al., 2018a) where $y_i < 1$ so the sign of $p_i - y_i$ for ground-truth class could still be positive, previous analytical methods fail to recover accurate label distribution. Limited progress has been made in this direction.

**Optimization-based gradient inversion attacks.** For optimization-based methods, the server initializes a data point $(x, y)$, minimizing the gradient matching distance between the ground-truth $(x^*, y^*)$ and the random data $(x, y)$ to push $(x, y)$ towards the ground-truth distribution $(x^*, y^*)$:

$$\arg \min_{(x^*, y^*)} \mathcal{D} \left( \nabla_\theta \mathcal{L}_\theta \left( x, y \right), \nabla_\theta \mathcal{L}_\theta \left( x^*, y^* \right) \right), \tag{1}$$

---

[2]Please refer to Section 3.1 for mathematical details.

where $\mathcal{D}$ represents the matching distance. Zhu et al. (2019) first adopt Euclidean distance to simultaneously optimize input data and labels. Subsequent works try different distance functions and add various regularization terms as real image priors for larger batches. Geiping et al. (2020) adopt cosine similarity as the distance function, and add total variance (Rudin et al., 1992); Wang et al. (2020) replace the $L_2$ distance by Gaussian kernel function, and apply different weights for different layer gradients; Yin et al. (2021) keep the original distance function, but borrow multiple regularization terms from DeepInversion (Yin et al., 2020) to strictly penalize the image towards realistic distribution; with similar logic, image recovery from Vision Transformers (Dosovitskiy et al., 2021) is also achieved (Hatamizadeh et al., 2022).

**Other image reconstruction methods.** Aono et al. (2017) first raise the bias attack to recover features from corresponding gradients in a biased fully-connected layer. Consider a fully-connected layer $\mathbf{z} = \mathbf{W}\mathbf{x} + \mathbf{b}$, $\frac{\partial \mathcal{L}(\mathbf{z},\mathbf{y})}{\partial \mathbf{W}_r} = \frac{\partial \mathcal{L}(\mathbf{z},\mathbf{y})}{\partial z_r} \times \frac{\partial z_r}{\partial \mathbf{W}_r} = \frac{\partial \mathcal{L}(\mathbf{z},\mathbf{y})}{\partial b_r}\mathbf{x}^{\mathrm{T}}$. If we know the gradients of the bias term, feature $\mathbf{x}$ could be retrieved from the division. Ma et al. (2023) also mention such methods as their theoretical intuition. After that, Fan et al. (2020) formulate image reconstruction as a linear equation system; Zhu & Blaschko (2020) initiate R-GAP, which solves a series of equations via Moore-Penrose pseudoinverse to get features from logits layer by layer, and finally to the original image. However, These two works depend more on the network architecture and parameters: data recovery methods will fail if the layer is rank-deficient. Pan et al. (2022) propose neuron exclusivity analysis to decouple batched image reconstruction into multiple single-image reconstructions. Balunovic et al. (2022) unify several gradient matching functions with a Bayesian framework and break several existing defenses in image reconstruction. Apart from these, Dang et al. (2021); Jeon et al. (2021) also include GAN (Goodfellow et al., 2020) for GIA, which requires extra data for GAN training.

## 3 LABEL INFERENCE FROM FULLY-CONNECTED NETWORKS

**Threat model.** Gradient inversion attack aims to reconstruct input data from gradient information in federated learning. Following the general setting where the honest-but-curious server wants to reconstruct private training data and labels (Li et al., 2022; Dang et al., 2021), it knows exactly the global model architectures as well as parameters $\mathbf{\Phi}_\theta$ (white-box attack), and could get full access to gradient uploads $g^* = \nabla_\theta \mathcal{L}_\theta (x^*, y^*)$ from clients. The server cannot actively change the training protocol to facilitate a better attack.

### 3.1 FROM RECOVERING A LABEL VECTOR TO OPTIMIZING A SCALAR $\lambda_r$

Considering the last fully-connected layer without the bias term $f : \mathbb{R}^I \to \mathbb{R}^C$ which outputs logits $\mathbf{z}$ from input $\mathbf{x}$ after activation, we have $\mathbf{z} = f(\mathbf{x}) = \mathbf{W}\mathbf{x}$ , where $I$ is the dimension of the input feature, $C$ represents the number of classes. Then the cross-entropy loss $\mathcal{L}(\mathbf{z}, \mathbf{y}) = -\sum_{i=1}^{C} y_i \log p_i$, where $\mathbf{y}$ refers to the label vector and $p_i$ is the $i$-th entry of post-softmax probability. We pick $\phi(\cdot)$ as the softmax function, then $p_i = \phi(\mathbf{z}, i) = \frac{e^{z_i}}{\sum_{t=1}^{C} e^{z_t}}$. Taking derivatives of row $r$ in weight matrix $\mathbf{W}$ from cross-entropy loss, we have

$$\frac{\partial \mathcal{L}(\mathbf{z}, \mathbf{y})}{\partial \mathbf{W}_r} = \frac{\partial \mathcal{L}(\mathbf{z}, \mathbf{y})}{\partial z_r} \cdot \frac{\partial z_r}{\partial \mathbf{W}_r}$$
$$= (p_r - y_r)\mathbf{x}^{\mathrm{T}}. \tag{2}$$

Apparently, the gradient of the last fully-connected layer is just a scaled input vector. If we do not know input $\mathbf{x}$, we are unable to solve each entry of $\mathbf{y}$ from Eqn. (2). However, label vector $\mathbf{y}$ could be regarded as a function of gradients $\mathbf{g}_r = \frac{\partial \mathcal{L}(\mathbf{z},\mathbf{y})}{\partial \mathbf{W}_r}$. For simplicity, we pick entry $i$ of label $\mathbf{y}$ to finish the equation[3]:

$$y_i = p_i - (p_i - y_i)$$
$$= \phi\left(f(\mathbf{x}^*)\right)_i - \frac{\mathbf{g}_i}{\mathbf{x}^*} \tag{3}$$
$$= \phi\left(f((p_r - y_r)^{-1}\mathbf{g}_r)\right)_i - (p_r - y_r)\frac{\mathbf{g}_i}{\mathbf{g}_r},$$

---

[3]Vector division makes sense here because the results of item-wise division are identical for every entry.

where $\mathbf{x}^*$ is the ground-truth input feature. If we replace the ground-truth input with $\lambda_r \mathbf{g_r}$, in which $\lambda_r$ is a non-zero scalar to be optimized, then the pseudo label could be written as

$$\hat{y}_i = \phi\left(f(\lambda_r \mathbf{g_r})\right)_i - \mathbf{g}_i/(\lambda_r \mathbf{g_r}). \tag{4}$$

When $\lambda_r^* = (p_r - y_r)^{-1}$, Eqn. (3) and Eqn. (4) are equivalent and then we get the ground-truth input feature $\lambda_r^* \mathbf{g_r}$ with label $\mathbf{y}^*$. With different $\lambda_r$, we could get different $\hat{\mathbf{y}}$ distribution. Based on this, our next goal is to design a target function to optimize this scalar $\lambda_r$. One point worth noting is that for all equations above, any random index $r$ satisfies the formula. Therefore, each non-zero $\mathbf{g}_r$ has its own $\lambda_r$. As long as we figure out one non-zero $\lambda_r$ for any index $r$, the ground-truth label could be successfully recovered.

## 3.2 Variance loss function

Here we take label smoothing and mixup techniques as our main focus in label augmentations. For label smoothing without access to the smoothing probability, the simplest way to regularize the label distribution is to minimize the variance of all label entries but the top one. Similarly, for mixup labels, we pick the variance loss of all but the top two entries.

$$\mathcal{L}_{label} = \frac{1}{C - \|\mathcal{S}\|} \sum_{i \notin \mathcal{S}}^{C} (\hat{y}_i - \mathbb{E}_{i \notin \mathcal{S}}(\hat{\mathbf{y}}))^2, \tag{5}$$

where $\mathcal{S}$ is the exclusion set containing the indexes of top items, $\|\mathcal{S}\|$ is the size of $\mathcal{S}$, and $\mathbb{E}$ means the expectation operator. During the process of picking $\lambda_r$, $\hat{\mathbf{y}}$ is changing so entries in $\mathcal{S}$ are not fixed. Ideally, when such a label is perfectly recovered, the variance loss of all-but-exclusion-set entries should reach the global minimum of $0$. It is worth highlighting that the target function is not unique, and other functions may also work. In experiments we find such simple variance does perform exceptionally well, and we will show the results in Section 4.

## 3.3 Searching for the global minimum

Based on Eqn. (5), the label recovery problem could be transformed to finding the global optimum in the $\mathcal{L}_{label}$ function. Here we first consider popular gradient-descent optimizers, such as Adam (Kingma & Ba, 2014), L-BFGS (Liu & Nocedal, 1989), etc. For most untrained networks, whose variance loss is as shown in Fig. 1(a), the global minimum is just the first local minimum (we start searching from $\pm1$), thus these gradient-descent-based algorithms could find the optimum easily and quickly. However, such a method suffers from multiple local minima and vanishing gradients, especially when retrieving labels from trained networks, whose $\mathcal{L}_{label}$ is shown in Fig. 1(b), 1(c).

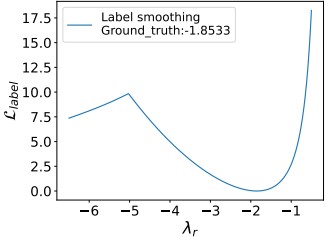 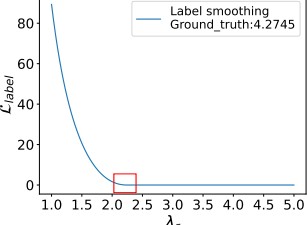 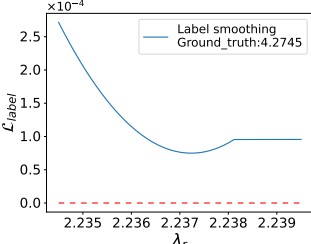

(a) $\mathcal{L}_{label}$ under untrained ResNet18. (b) $\mathcal{L}_{label}$ under trained ResNet18. (c) A detailed illustration of the red-
Clearly the first local minimum is the The first local minimum is not the box area in Fig. 1(b). $\mathcal{L}_{label}$ does
global one. global one. not get zero.

Figure 1: Variance loss under untrained and pretrained ResNet18. The gradient-based optimizer may find local minima at approximately 2.2372 while the ground-truth is 4.2745.

Under such circumstances, Particle Swarm Optimization (Kennedy & Eberhart, 1995) is a better choice to find the global minimum, for it will not suffer from vanishing gradients and local minima. In experiments we first use L-BFGS to try to find the global optimum, then PSO if L-BFGS fails.

### 3.4 THE NECESSARY CONDITION IN LABEL RECOVERY ALGORITHMS

One important conclusion derived from our label recovery problem is that the ground-truth label conditioning gradient information is a function of the scalar $\lambda_r$. Faced with such uncertainty, we only have two handles for $\lambda_r$ picking:

- All entries of the valid label should sum to 1.
- All entries of the valid label are and only could be non-negative.

**Theorem 1.** *For any $\lambda_r$ and gradient information $\mathbf{g_r}$, all entries of the corresponding pseudo label $\hat{\mathbf{y}}$ sum to 1.*

The proof is attached in Appendix A. Therefore, we could only focus on the second handle. However, this non-negative principle is not always useful[4], either. If we only get access to the gradient information, there would be a range for $\lambda_r$ in which any $\lambda_r$ could produce a reasonable label. Here is an intuitive example of label smoothing augmentation.

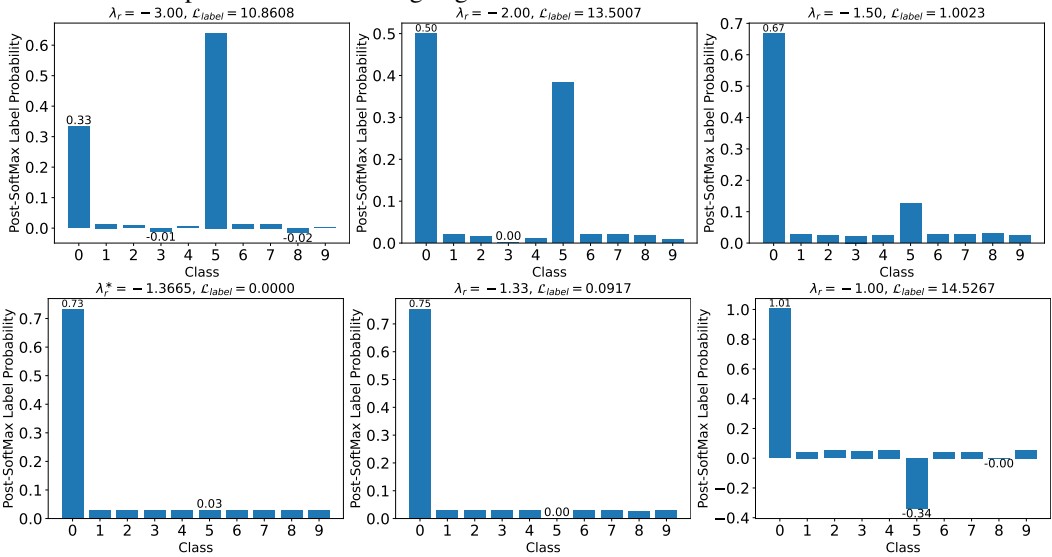

Figure 2: Recovered label distribution and variance loss given same gradient information. When we alter the scalar $\lambda_r$, the class probability would vary. If we have no more information about label distribution, all labels except the left-top and right-bottom ones could be the right labels generating exactly the same gradients.

As shown in Fig. 2, at least any $\lambda_r$ ranging from $-2.00$ to $-1.33$ (where one entry in probability distribution almost reaches 0) would generate labels where each entry is positive and sums to 1. That is why we need the variance loss to supervise the label distribution. Based on this, **label distribution feature is a necessary condition as guidance to pick one specific label from the series controlled by $\lambda_r$.** Label distribution feature refers to features that could be utilized to design the target function for label recovery. Such a theory is also applicable to mixup labels. Actually, our algorithm could be easily adapted to other augmented labels, as long as we know the special feature of that kind of label distribution and then design new loss to guide $\lambda_r$ picking. Otherwise, these two handles without prior knowledge of label distribution are too weak to guarantee an accurate recovery.

## 4 EXPERIMENTS

### 4.1 LABEL RECOVERY ACCURACY

We first test the label recovery accuracy with CIFAR-100 (Krizhevsky, 2009), Flowers-17 (Nilsback & Zisserman, 2006) and ImageNet (Russakovsky et al., 2015) on ResNet50 (He et al., 2016) network.

---

[4]Under some circumstances this handle could help directly optimize the ground-truth scalar. More details could be found in Appendix C.

In label smoothing settings, we randomly pick 1000 images from the testset of each dataset (for Flowers-17 we pick images from the whole dataset). To test the robustness of our algorithm, we randomly pick the smoothing probability from $U(0, 0.5)$ for every sample. Besides, we also consider the one-hot setting, which is a special case for label smoothing when the smoothing probability is 0, to testify to the algorithm performance. In the mixup setting, we also randomly pick 1000 image couples in different classes with the probability from $U(0, 1)$.

Table 1: Experiments for label recovery accuracy.

| Label | Accuracy (%) | | |
|---|---|---|---|
| Augmentation | CIFAR-100 | Flowers-17 | ImageNet |
| Label smoothing (L) | 99.2 | 99.1 | **100.0** |
| Mixup (M) | **100.0** | 97.6 | **100.0** |
| One-hot | **100.0** | 95.6 | 99.9 |
| iDLG[5] | **100.0** | 95.6 | 99.9 |

As shown in Table 1, firstly our proposed algorithm achieves satisfying recovery accuracy of above 95% for both label augmentations. Besides, our method also achieves identical performance on the one-hot setting as the previous iDLG in multiple datasets, demonstrating its compatibility. To further test the performance of the proposed label recovery algorithm, we then focus on CIFAR-10 and Flowers-17 to execute more intense experiments. For both datasets, we pick two different networks and test our algorithm both on pretrained and random-initialized networks, as shown in Table 2.

Table 2: Intense experiments on the performance of label recovery algorithm. $\mathcal{L}_r$ refers to the sum of $L_1$ loss of every entry in recovered labels, if not otherwise specified.

| CIFAR-10 | | | | Flowers-17 | | | |
|---|---|---|---|---|---|---|---|
| Network | Aug. | Acc. (%) | Avg. $\mathcal{L}_r$ | Network | Aug. | Acc. (%) | Avg. $\mathcal{L}_r$ |
| Untrained | L | 99.7 | $5.32\times10^{-5}$ | Untrained | L | 94.6 | $2.22\times10^{-6}$ |
| LeNet | M | 99.7 | $3.62\times10^{-5}$ | AlexNet | M | 88.3 | $9.64\times10^{-6}$ |
| Trained | L | 99.9 | $2.39\times10^{-4}$ | Trained | L | 98.7 | $1.09\times10^{-4}$ |
| LeNet | M | **100.0** | $1.51\times10^{-4}$ | AlexNet | M | 98.9 | $6.50\times10^{-8}$ |
| Untrained | L | **100.0** | $8.78\times10^{-5}$ | Untrained | L | **100.0** | $6.02\times10^{-6}$ |
| ResNet18 | M | **100.0** | $7.50\times10^{-5}$ | ResNet18 | M | **100.0** | $2.92\times10^{-5}$ |
| Trained | L | 99.2 | $3.17\times10^{-7}$ | Trained | L | 99.1 | $7.36\times10^{-5}$ |
| ResNet18 | M | 99.0 | $3.94\times10^{-6}$ | ResNet18 | M | 99.9 | $2.30\times10^{-4}$ |

Our algorithm could achieve above 95% accuracy on multiple datasets under various training conditions, which demonstrates its robustness. In addition, it does not experience an obvious accuracy drop when the networks are pretrained with mixup or label smoothing techniques. Failure analysis for corner cases is in Appendix B.

## 4.2 IMAGE RECONSTRUCTION FOR FULLY-CONNECTED NETWORKS

As mentioned in R-GAP (Zhu & Blaschko, 2020), simply canceling the bias term of every layer would disable previous bias attack (Aono et al., 2017; Geiping et al., 2020). With the proposed label recovery algorithm, the ground-truth feature of the last layer could directly be recovered from the cross-entropy loss of multi-class classification tasks, therefore reconstructing image analytically from fully-connected networks is not limited to a theoretical level.

Consider one fully-connected layer with input $\mathbf{x}$, weight $\mathbf{W}$ and output $\mathbf{z} = \mathbf{W}\mathbf{x}$ before activation, then we have

$$\frac{\partial \mathcal{L}}{\partial \mathbf{x}} = \mathbf{W}^{\mathrm{T}} \cdot \frac{\partial \mathcal{L}}{\partial \mathbf{z}} \tag{6}$$

---

[5]iDLG (Zhao et al., 2020) could only handle hard labels and therefore serves as a comparison with proposed variance-optimizing methods in a one-hot setting. From experiments, we find that iDLG cannot guarantee 100% accuracy, which may contradict previous consensus. A detailed explanation is attached in Appendix B.3.

$$\mathbf{x}_r = \left(\frac{\partial \mathcal{L}}{\partial \mathbf{W}}\right)_{i,r} \cdot \left(\frac{\partial \mathcal{L}}{\partial \mathbf{z}}\right)_i^{-1}, \tag{7}$$

where we need to pick $i$ to make $\left(\frac{\partial \mathcal{L}}{\partial \mathbf{W}}\right)_{i,r}$ non-zero. If the activation function is ReLU (Glorot et al., 2011), then $\frac{\partial \mathcal{L}}{\partial \mathbf{x}}$ is identical to $\left(\frac{\partial \mathcal{L}}{\partial \mathbf{z}}\right)$ of the previous layer. For other activation functions the logic is similar and we only need to multiply a scalar.

Thus, after recovering this layer's input feature and gradients of the previous layer's output feature, we could recurrently reconstruct the input feature of previous layers. Such a method successfully avoids solving the output feature of each layer, especially when the activation function is not strictly monotonic. As shown in Fig. 3, our analytical-based algorithm could reconstruct high-quality images with label augmentations, which outperforms DLG and IG with a shorter running time.

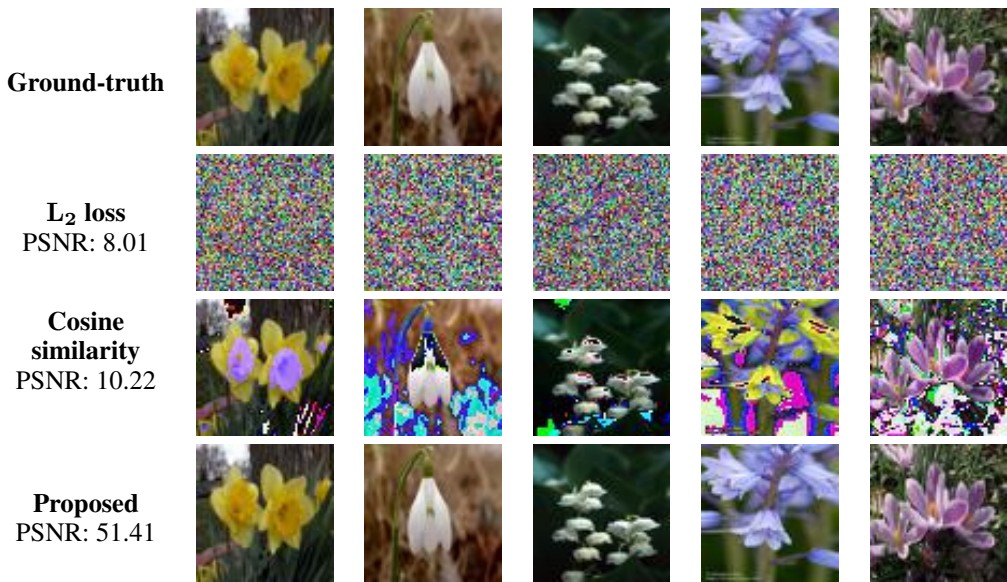

Figure 3: Image reconstruction comparisons on FCN-4 network. Images with label smoothing are compressed to $64 \times 64$ due to CUDA memory limitation in optimization-based methods. For more samples with mixup augmentation please refer to Appendix D.

### 4.3 RECONSTRUCTION FOR CNNs

In this part, we further discuss the benefits accurate label recovery brings to image reconstruction. We pick widely used $L_2$ (Zhu et al., 2019; Yin et al., 2021) loss and cosine similarity (Geiping et al., 2020) as the loss function, evaluating image reconstruction quality with the precisely recovered label (Ours), ground-truth label (GT), one-hot label (iDLG), randomly initialized label (DLG) and our recovered label as initialization (DLG+Ours), respectively. Experiments are finished on LeNet (LeCun et al., 1998) architecture with CIFAR-10 validation dataset. For DLG and DLG+Ours, labels are involved in the optimization process, while for the other three settings, we fix the labels and only optimize the input image. In the mixup setting, we have 45 different class tuples and for each tuple, we randomly pick one image in the corresponding class to mix with the probability sampling from $U(0, 1)$. In the label smoothing setting, for each class, we pick 3 random images and smooth every label with probability sampling from $U(0, 0.5)$. All experiments are repeated 10 times. Evaluation metrics are widely used Peak Signal-Noise Ratio (PSNR), Structural Similarity (SSIM), and Learned Perceptual Image Patch Similarity (LPIPS) (Zhang et al., 2018b). Detailed results are shown in Table 3[6].

**Proposed method outperforms previous label recovery methods with a large margin.** For augmented labels, DLG method fails to recover labels as accurately as the proposed method. The optimization method is unstable, so for different initializations the $\mathcal{L}_r$ varies. Besides, For $L_2$ loss,

---

[6]A prior version of this paper had a bug in the image reconstruction implementation, which is unrelated to our label recovery algorithm. After corrections, all conclusions still hold.

Table 3: Image reconstruction with label loss under various settings. $L_2$ loss and Cosine similarity refer to two ways of calculating matching distance $\mathcal{D}$ in Eqn. 1.

| PSNR↑ SSIM↑ LPIPS↓ | Match function | Trained | GT | iDLG | DLG | Ours | DLG + Ours | $\mathcal{L}_r$ | |
| --- | --- | --- | --- | --- | --- | --- | --- | --- | --- |
| | | | | | | | | DLG | Ours |
| Mixup | $L_2$ loss | ✗ | **28.27**
0.640
0.141 | 10.70
0.099
0.355 | 24.26
0.553
0.172 | 28.24
0.644
0.138 | 27.81
**0.659**
**0.130** | 0.44 | $5\times10^{-6}$ |
| | $L_2$ loss | ✓ | 19.28
0.628
**0.062** | 9.34
0.020
0.394 | 16.93
0.500
0.113 | **19.29**
**0.628**
0.063 | 17.30
0.519
0.102 | 0.15 | $8\times10^{-6}$ |
| | Cosine similarity | ✗ | 25.76
0.858
0.033 | 23.01
0.745
0.053 | 26.45
0.881
0.027 | 25.81
0.859
0.033 | **26.51**
**0.882**
**0.027** | 0.11 | $5\times10^{-6}$ |
| | Cosine similarity | ✓ | 24.56
0.741
0.046 | 15.51
0.356
0.173 | 22.71
0.688
0.053 | 24.67
0.746
0.044 | **24.84**
**0.758**
**0.040** | 0.05 | $8\times10^{-6}$ |
| Label smoothing | $L_2$ loss | ✗ | **21.33**
0.587
0.150 | 10.64
0.129
0.351 | 18.98
0.492
0.192 | 20.94
0.584
**0.149** | 21.00
**0.590**
0.149 | 0.37 | $4\times10^{-6}$ |
| | $L_2$ loss | ✓ | 11.70
0.267
0.250 | 8.72
0.029
0.401 | **11.86**
**0.283**
**0.236** | 11.61
0.260
0.259 | 11.47
0.244
0.264 | 0.36 | $8\times10^{-6}$ |
| | Cosine similarity | ✗ | 23.13
0.820
0.034 | 22.89
0.807
0.032 | 24.09
0.865
0.023 | 23.06
0.818
0.035 | **24.26**
**0.868**
**0.022** | 0.20 | $4\times10^{-6}$ |
| | Cosine similarity | ✓ | 18.99
0.615
0.095 | 17.40
0.558
0.121 | 18.56
0.614
0.088 | 18.98
0.614
0.095 | **20.42**
**0.691**
**0.069** | 0.17 | $8\times10^{-6}$ |

one-hot labels derived from iDLG also fail to provide satisfying image reconstruction quality.

**Proposed method successfully reaches image reconstruction quality as known labels.** In all experiments, our proposed method recovers high-quality labels with $\mathcal{L}_r$ under $8.5 \times 10^{-6}$. Under all experimental settings, recovering images with recovered labels is almost identical to recovering with ground-truth labels in all metrics, demonstrating the effectiveness of our algorithm.

**Proposed method could serve as a good initialization for cosine similarity loss.** For cosine similarity loss, under the untrained setting, optimizing an image-label tuple simultaneously could even outperform the image quality that is recovered from the ground-truth label. Even though it is a special character of cosine similarity loss itself, we do find that optimizing the image-label tuple with our recovered label as an initialization could increase image reconstruction quality. More importantly, such a margin would be enlarged under the pretrained setting, demonstrating the value of our algorithm more convincingly.

## 5 EXTENSION STUDIES

### 5.1 RECONSTRUCTING IMAGE FROM GRADIENTS AND FEATURES

Given that our label recovery algorithm could recover features and labels without the bias term simultaneously, we could include last-layer features in the matching loss to evaluate the profits our algorithm brings to gradient inversion attacks:

$$\mathcal{L}_{match} = \sum_{i=0}^{N-2} \mathcal{D}(g_i, g_i{}^*) + \alpha\mathcal{D}(g_{N-1}, g_{N-1}{}^*) + \beta\mathcal{D}(f_{N-1}, f_{N-1}{}^*), \tag{8}$$

where $\alpha$ and $\beta$ refer to the weights of last-layer gradient loss and feature loss. The remaining settings are identical as in Section 4.3. As shown in Table 4, the last-layer feature does improve the image reconstruction quality and $\beta = 2$ has the best image reconstruction results under most settings. Besides, considering features and gradients of the same layer in the loss function would squeeze the final reconstruction quality. Feature information is more direct, and therefore deserves priority with higher weights.

Table 4: Image reconstruction with feature loss considered into gradient matching loss.

| PSNR↑ SSIM↑ LPIPS↓ | Match function | $\alpha = 1$ $\beta = 0$ (Baseline) | $\alpha = 0$ $\beta = 1$ | $\alpha = 1$ $\beta = 1$ | $\alpha = 0$ $\beta = 2$ | $\alpha = 2$ $\beta = 2$ |
|---|---|---|---|---|---|---|
| Mixup | $L_2$ loss | 29.04 0.677 0.124 | 34.09 0.763 0.089 | 29.28 0.686 0.120 | **34.18** **0.762** **0.088** | 29.53 0.691 0.116 |
| | Cosine similarity | 25.71 0.856 0.034 | 28.75 0.918 **0.015** | 25.89 0.857 0.033 | **28.81** **0.918** 0.015 | 25.92 0.858 0.032 |
| Label smoothing | $L_2$ loss | 20.49 0.535 0.176 | **24.96** **0.690** **0.114** | 21.00 0.578 0.162 | 24.65 0.683 0.117 | 20.80 0.559 0.167 |
| | Cosine similarity | 23.02 0.817 0.035 | 25.79 0.885 0.017 | 23.27 0.824 0.032 | **25.85** **0.887** **0.016** | 23.34 0.824 0.032 |

## 5.2 DIFFERENTIAL PRIVACY

The accuracy of label recovery is based on accurate ground-truth gradients. Therefore, in this part, we consider the impact that widely used differential privacy techniques may have on the label recovery quality. To evaluate, we follow previous work (Zhu et al., 2019) to add Gaussian and Laplace disturbance with variance from $10^{-4}$ to $10^{-1}$ and central 0 to last-layer gradients. In all experiments, we pick 100 samples in the CIFAR-10 validation dataset from 10 classes with label smoothing augmentation under untrained ResNet18. Only the PSO algorithm is utilized to find the optimum.

Table 5: Label recovery accuracy and $\mathcal{L}_s$ under Gaussian and Laplace disturbance. $\mathcal{L}_s$ refer to the difference between $\lambda_r$ and $\lambda_r^*$.

| Noise | Gaussian | | | | Laplace | | | |
|---|---|---|---|---|---|---|---|---|
| Variance | $10^{-4}$ | $10^{-3}$ | $10^{-2}$ | $10^{-1}$ | $10^{-4}$ | $10^{-3}$ | $10^{-2}$ | $10^{-1}$ |
| $\mathcal{L}_s$ | 1.02e-4 | 1.13e-3 | 1.14e-2 | 5.82e-1 | 1.61e-4 | 8.07e-4 | 7.95e-3 | 7.95e-1 |
| Acc. (%) | 100 | 100 | 100 | 45 | 100 | 100 | 100 | 36 |

For both Gaussian and Laplace disturbance, accuracy only drops harshly when variance reaches $10^{-1}$. That means disturbance with such variance raises the variance loss of the original optimum scalar to a level greater than other local minima, causing minimum exchange, while disturbance variance ranging from $10^{-4}$ to $10^{-2}$ only adds a slight noise to the optimal scalar.

## 6 CONCLUSION

This work proposes the first label recovery algorithm to analytically retrieve augmented labels as well as last-layer input features from gradients. We analyze the limitations of previous single-image label recovery methods, provide a necessary condition for label recovery, and design the first algorithm to recover high-quality images from unbiased fully-connected networks under multi-class classification tasks. Extensive experiments have proved the recovery accuracy and quality, together with the benefits of following image reconstruction on both fully-connected networks and CNNs. We hope augmented labels, together with other real-world settings in GIA, could attract more attention.

ACKNOWLEDGEMENT

We thank Aijing Yu and Jiyang Guan in CRIPAC for their feedback on our early drafts. Besides, we also would like to present our appreciation to the anonymous reviewers for their constructive suggestions, with which we could make our expression more fluent and informative. This work was supported by the National Natural Science Foundation of China (No. 62276256, No. U21B2045, No. U20A20223, No. 32341009), the Beijing Nova Program under Grant Z211100002121108 and Young Elite Scientists Sponsorship Program by CAST.

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

## A   PROOF FOR THEOREM 1

**Theorem 1.** *For any $\lambda_r$ and gradient information $\mathbf{g_r}$, all entries of the corresponding pseudo label $\hat{\mathbf{y}}$ sum to 1.*

*Proof.* According to Eqn. 4, $\hat{y}_i = \phi\left(f(\lambda_r \mathbf{g}_r)\right)_i - \mathbf{g}_i/(\lambda_r \mathbf{g}_r)$. Therefore, we have

$$\sum_{i=1}^{C} \hat{y}_i = \sum_{i=1}^{C} \phi\left(f(\lambda_r \mathbf{g}_r)\right)_i - \mathbf{g}_i/(\lambda_r \mathbf{g}_r) \tag{9}$$

Because $\phi$ refers to the softmax function, we have

$$\sum_{i=1}^{C} \phi\left(f(\lambda_r \mathbf{g}_r)\right)_i = 1 \tag{10}$$

Therefore, we only need to prove $\sum\limits_{i=1}^{C} \mathbf{g}_i/(\lambda_r \mathbf{g}_r) = 0$. From Eqn. 2, we get $\mathbf{g}_i = (p_i - y_i)\mathbf{x}^{\mathrm{T}}$, so

$$\sum_{i=1}^{C} \mathbf{g}_i = \left(\sum_{i=1}^{C} p_i - \sum_{i=1}^{C} y_i\right) \mathbf{x}^{\mathrm{T}} = \mathbf{0} \tag{11}$$

Consequently, $\sum\limits_{i=1}^{C} \mathbf{g}_i/(\lambda_r \mathbf{g}_r) = 0$. The theorem was proved.

## B   FAILURE ANALYSIS

### B.1   DISTURBANCE OF LOCAL MINIMA

The proposed label recovery algorithm focuses on finding the global minimum while avoiding local minima. However, in real searching processes, none of the algorithms could find the exact minimal point with a loss value equal to 0. Therefore, we increase variance loss by 1000 times and regard optimal points with loss below $10^{-9}$ as potential solutions. Under extreme conditions, some local minima also reach such a low variance loss, causing recovery failure. This failure could be resolved by lowering the loss bar and increasing the population of particles in PSO, which would take more time.

### B.2   OUT OF BOUND

Consider logit $\mathbf{z}$ has its maximal entry $z_{max}$ and minimal entry $z_{min}$ ($max$ and $min$ are subscript of entries). If scalar $\lambda_r \to +\infty$ or $\lambda_r \to -\infty$, $SoftMax(\mathbf{z})$ would show one-hot distribution: $p_{max} \to 1$ or $p_{min} \to 1$, both of whose variance function approach 0. Therefore, if we keep searching without bounds, there exists $\delta$ so that for any $\lambda_r$, if $\lambda_r > \delta$, then $\mathcal{L}_{label} < 10^{-9}$.

Because ground-truth $\lambda_r{}^* = (p_r - y_r)^{-1}$, for trained networks, $p_r$ and $y_r$ could get close so $\lambda_r{}^*$ would be large enough to exceed bounds. This phenomenon also causes recovery failure. Extending searching range could solve such a problem, but in experiments, it consumes more time. So it is also a time-accuracy trade-off.

### B.3   ONE-HOT PROBABILITY

This is a systematic error when ground-truth labels are in one-hot type, especially referring to the experiment in table 1, which cannot be solved by any algorithm currently, including previous iDLG (Zhao et al., 2020). Consider input $\mathbf{x}$ generates one-hot probability after the softmax layer, and the label is also in one-hot type. Assign the entry with value 1 is $i$ and $j$, meaning $p_i = y_j = 1$, then we have two situations:

- $i = j$, $\mathcal{L}_{CE} = 0$, $\mathbf{g} = \mathbf{0}$
- $i \neq j$, $\mathcal{L}_{CE} > 0$, $\mathbf{g}_i = -\mathbf{g}_j$

For the first situation, the model predicts the input with 100% accuracy and strict 0 loss, so ground-truth $\lambda_r{}^*$ is infinite, causing zero-division error. This is easy to handle because we already know that the model output is identical to the ground-truth label. In code implementations, we simply pass the optimization process, directly outputting model prediction as recovered one-hot labels.

For the second situation, it gets complicated. Originally, the ground-truth $\lambda_r{}^*=(p_j - y_j)^{-1} = -1$. However, the algorithm may take $\lambda_r = 1$ as the ground-truth because it did produce a one-hot label with zero all-but-top-one variance, even though the entry with value 1 is not correct. In iDLG (Zhao et al., 2020), theoretically we hold that in the gradient information matrix, the entry with a negative sign(e.g., $p_j - y_j < 0$) is the ground-truth label. But in fact, the negative sign is figured out by checking which entry has a gradient tensor whose sign is different from all other gradients, because gradient values themselves do not share the same sign under most circumstances. For this specific situation, only two entries of $\mathbf{g}$ are non-zero, so we cannot decide which one is $j$. That is how the sign indicator fails. Such mathematical relation only comes out in corner cases when the model is randomly initialized and predicts some inputs with extremely high probabilities. This explains why our proposed method shares identical non-perfect accuracy with iDLG, which also demonstrates the correctness and compatibility of such a method.

Such one-hot prediction also influences the mixup setting, in which $i$ and $j$ are perceived as entries involved in mixing. When networks are properly trained, even with only a few epochs, such extreme one-hot predictions will not be generated from models anymore, implying satisfying label recovery accuracy.

## C  EXAMPLES OF OTHER TARGET LOSS FUNCTIONS

- $\mathcal{L}_{label} = \frac{1}{C-||\mathcal{S}||} \sum\limits_{i \notin \mathcal{S}}^{C} \hat{\boldsymbol{y}}_i^2$

- $\mathcal{L}_{label}(\lambda_r) = \sum\limits_{i=1}^{C} ||\hat{y_i}|| - 1$

For the first loss function, we design intuitive experiments on ResNet18 with CIFAR-10 datasets in a label smoothing setting. We get 100% accuracy on the untrained network and 97.78% on the trained network. As stated in the main literature, the target loss function is not unique.

The second function is a loss to penalize negative entries in the pseudo label. Under some circumstances, especially one-hot or mixup labels where multiple ground-truth entries are 0, the change of each entry tends to be in a different direction during the optimization process. Therefore, it is possible that only the ground-truth label has all non-negative entries, implying that this loss function may only have one global minimum 0. However, such property is not guaranteed. To make sure a successful recovery, under most circumstances we need another necessary condition to guide target function designing.

## D  FCN NETWORK RECOVERY VISUALIZATION

In this part, we display all recovered 100 images from label smoothing and mixup augmentation, respectively. The reconstruction results from FCN are satisfying enough so we do not bother adding ground-truth images as comparisons. Experimental results are shown in Table 6.

Table 6: Average image reconstruction performance overall randomly picked data.

| Augmentation | SSIM | PSNR | LPIPS |
|---|---|---|---|
| Label smoothing | 0.999 | 51.30 | 0.001 |
| Mixup | 1.000 | 66.80 | 0.045 |

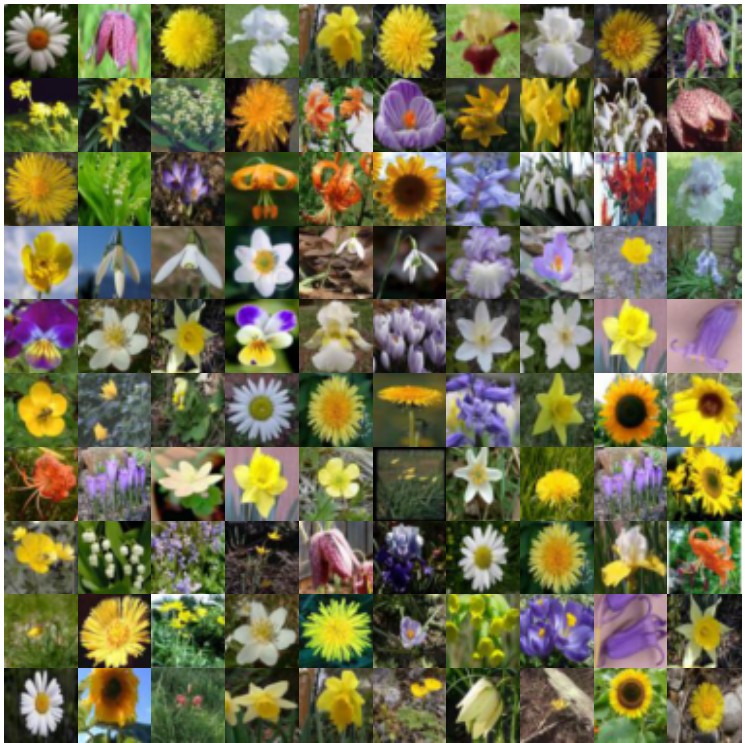

Figure 4: Recovered images with label smoothing augmentations from FCN-4. All 100 images are randomly picked with unknown smoothing probability.

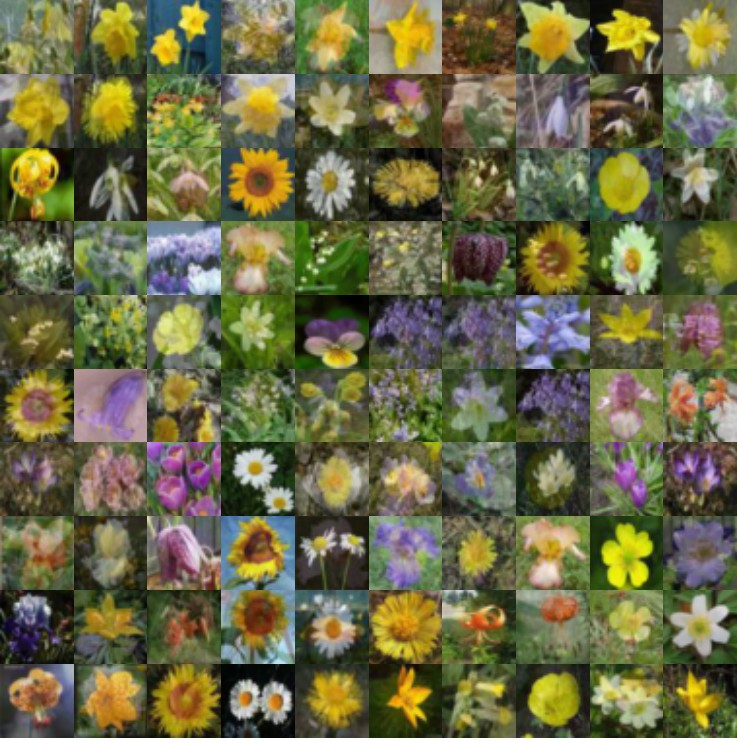

Figure 5: Recovered images with mixup augmentations from FCN-4. All 100 mixed images are randomly picked with unknown mixup probability.

# E  PSEUDOCODE FOR LABEL RECOVERY ALGORITHM

Here we present a big picture for our label recovery algorithms with gradient-descent based optimizer and PSO optimizer, as shown in Algorithm 1 and Algorithm 2. Full details can be checked in our repository.

---

**Algorithm 1:** Gradient-descent-based label recovery algorithm

---

**Input:** $\mathbf{g_m}$: one entry of last-layer gradients with maximal absolute sum; $lr$: learning rate; $initial$: initial points; $coe$: coefficient; $bound$: searching upper bound; $iteration$: maximum optimization iterations; $func$: input feature, output post-softmax probability; $var$: variance loss function for label distribution.

**Output:** Scalar $\lambda_r$

$\lambda_r = initial$
$\mathbf{g} = coe \times \mathbf{g_m}$
Label=$func(\lambda_r \times \mathbf{g})$
Loss=$var$(Label)
Opt=$optimizer$(Loss, $lr = lr$)
**for** $i$ **in range** ($iteration$ // 2): **do**
  Label= $func(\lambda_r \times \mathbf{g})$
  Loss= $var$(Label)
  **if** Loss<$1e - 9$: **then**
    | **return** $\lambda_r$
  **end**
  **else if** $\lambda_r > bound$ **then**
    | **break**;
  **end**
  Loss.*backward()*
  Opt.*step()*
**end**
$\lambda_r = -1 \times initial$ // Restart from the negative side if failed.
Label=$func(\lambda_r \times \mathbf{g})$
Loss=$var$(Label)
Opt=$optimizer$(Loss, $lr = lr$)
**for** $i$ **in range** ($iteration$ // 2, $iteration$): **do**
  Label= $func(\lambda_r \times \mathbf{g})$
  Loss= $var$(Label)
  **if** Loss<$1e - 9$: **then**
    | **return** $\lambda_r$
  **end**
  **else if** $\lambda_r > bound$ **then**
    | **return** $-1$;
  **end**
  Loss.*backward()*
  Opt.*step()*
**end**
**return** $-1$

---

For L-BFGS, we set the learning rate=0.5, bound=100, iteration=200, coefficient=4, and initial=1. For PSO, we set initial=1, pop=200, max_ iter=30 by default. Enlarging the particle population as well as the iterations could enhance accuracy with the sacrifice of running time. Coefficient and initial could be fine-tuned for better performance.

---

**Algorithm 2:** PSO-based label recovery algorithm.

---

**Input:** $\mathbf{g_m}$: one entry of last-layer gradients with maximal absolute sum; $initial$: initial points; $coe$: coefficient; $bound$: searching upper bound; $iteration$: maximum optimization iterations; $func$: input feature, output post-softmax probability; $var$: variance loss function for label distribution; $interval$: searching range for each step.

**Output:** Scalar $\lambda_r$

$\mathbf{g}=coe \times \mathbf{g_m}$
Label=$func(\lambda_r \times \mathbf{g})$
Loss=$var$(Label)
lower_ bound=$initial$; upper_ bound=lower_ bound+$interval$
**while** upper_ bound<$bound$ **do**
  pso=PSO(Loss, lower_ bound=lower_ bound$-0.3$, upper_ bound=upper_ bound, pop=pop, max_ iter=max_ iter)
  pso.*run()*
  $\lambda_r$=pso.*bestx*
  Loss=pso.*besty* // pso.bestx and pso.besty save the minimum data point.
  **if** Loss<$1e-9$: **then**
  | **return** $\lambda_r$
  **end**
  **else**
  | pso=PSO(Loss, lower_ bound=-upper_ bound$-0.3$, upper_ bound= $-$lower_ bound, pop=pop, max_ iter=max_ iter)
  | pso.*run()*
  **end**
  $\lambda_r$=pso.*bestx*
  Loss=pso.*besty*
  **if** Loss<$1e-9$: **then**
  | **return** $\lambda_r$
  **end**
  lower_ bound=upper_ bound; upper_ bound=upper_ bound+$interval$
**end**
**return** $-1$

---

## F  MORE DATA FOR SECTION 5.2

Here we also note down the $\mathcal{L}_r$ when faced with noisy gradients. As mentioned in Section 4.1, $\mathcal{L}_r$ refers to the sum of $L_1$ loss from every entry of recovered labels. Results are shown in Fig. 6.

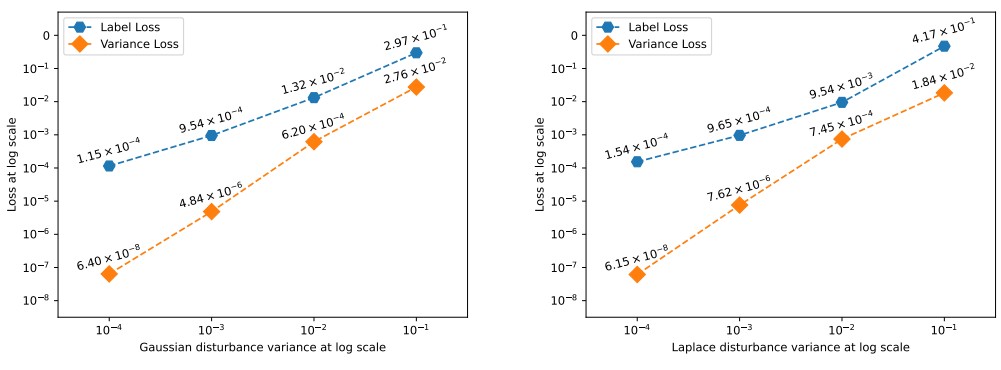

Figure 6: Loss comparison under different disturbance intensity

## G    A FULL VERSION OF FIG. 1

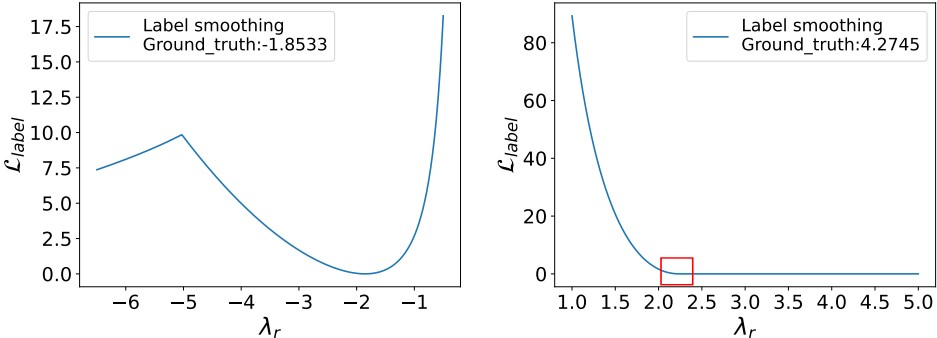

(a) $\mathcal{L}_{label}$ under untrained ResNet18. Clearly the first local minimum is the global one.

(b) $\mathcal{L}_{label}$ under trained ResNet18. The first local minimum is not the global one.

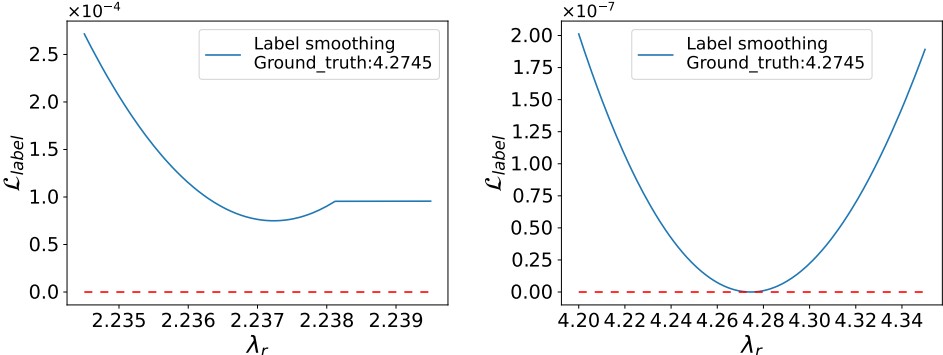

(c) A detailed illustration of the red-box area in Fig. 1(b). $\mathcal{L}_{label}$ does not get zero.

(d) A detailed illustration of the global minimum area in Fig. 1(b). $\mathcal{L}_{label}$ reaches zero.

Figure 7: Variance loss under untrained and pretrained ResNet18. The gradient-based optimizer may find local minima at approximately 2.2372 while the ground truth is 4.2745.

