# OpenReview forum: "Towards Eliminating Hard Label Constraints in Gradient Inversion Attacks"
_ICLR.cc/2024/Conference — ICLR 2024 poster_

### Official Review · Reviewer_wnMY · 2023-10-29

**Soundness:** 3 good
**Presentation:** 3 good
**Contribution:** 3 good
**Rating:** 8
**Confidence:** 4

**Summary:**

This paper proposes a method for gradient inversion attack with relaxation from hard labels to soft labels. The proposed method is based on a variance loss function and corresponding analysis is presented.

**Strengths:**

1. This paper identifies an interesting gap for gradient inversion attack, i.e., the hard label constraints, and proposes a soft label recovery method that is closer to realistic scenarios.

2. Although the proposed method is simple, it is intuitive and effective. Its simplicity also adds value to its applicability.

3. Effectiveness is shown by the experimental results quantitatively and qualitatively.

4. Code is provided in the supplementary.

**Weaknesses:**

My major question is about the assumption made regarding the label format. Does the proposed method assume the label format is known in advance or not?

- First, does the method assume it is known whether a hard label or soft label is used?
- Second, does the method assume it is known whether label smoothing or mixup is used as the format for the soft label?

If they are all known, I am curious about other versions of the results in Table 3 when the label format is unknown, corresponding to the above two settings. One setting is that we don't know whether the label is hard or soft. The other setting is that we don't know the specific format of the soft label. I think these two settings are closer to the real-world cases.

Minors:
- In Figure 2, it is better to also zoom in and visualize the global minimum similarly.
- In Table 2 caption, experiences > experiments.

**Questions:**

Please refer to the Weaknesses. I would appreciate comments on the assumption of the label formats and additional results if applicable.

---

> ### Author Response · Authors · 2023-11-14
>
> Thanks for the detailed review and insightful questions! The paper is revised, and due to the 9-page limitation, we added the global minimum figure in Section H.
>
> Firstly, **we do assume that we know the label type in detail.** For our algorithm, the two settings are the same, because only knowing the label is soft without the specific type cannot help design the loss function to replace the variance, and one-hot labels, a.k.a. hard labels, are just a special case of label smoothing: the smoothing factor is 0. I would prefer to split the setting in this way:
>
> 1. we **do not know anything** about the label.
> 2. we do know the label may belong to mixup, label smoothing, or other types. we **have a finite set with all possibilities**.
>
> So, the question is: "Could the algorithm recover images under the two circumstances mentioned above?" It is a little complicated. As stated in Section 3.4, if we know nothing about the label, then we only get the first condition: All entries of the valid label are and only could be non-negative. From this condition, we could only design this loss to replace the variance:
>
> $$\mathcal{L}_{label}(\lambda_r)=\sum\limits_{i=1}^{C}{|\hat{y_i}|}-1$$
>
> For label smoothing, We first executed label recovery on ResNet18 and LeNet for 200 samples each, finding that both recovery accuracies are 0 (the loss is always above the bound).
>
> We then executed experiments on image reconstruction with the $L_2$ match function. we randomly picked 2 images for each class and repeated 3 times. Other settings are kept identical to Tabel 3. The results are shown below.
> |      | GT label |  DLG  |  Ours  | DLG+Ours |
> |:----:|:--------:|:-----:|:-----:|:--------:|
> | PSNR |  14.61   | 9.49  | 12.38 |   9.76   |
> | SSIM |  0.523   | 0.144 | 0.367 |  0.148   |
>
> Clearly, inaccurate labels cause a performance drop in image reconstructions. The `PSNR` and `SSIM` in the `Ours` column are lower than those in the `GT label`. It is because without variance supervision, as shown in Fig. 2, **multiple labels could all generate the same gradients**. Under such circumstances, the algorithm may randomly stay at one label whose entries are all above zero.
> Here is a corner example. The ground-truth label is`[0.031, 0.031, 0.031, 0.031, 0.031, 0.031, 0.031, 0.717, 0.031, 0.031]`, but the recovered is `[0.086, 0.125, 0.147, 0.080, 0.139, 0.078, 0.089, 0.100, 0.080, 0.077]`, which diverges a lot. Distribution change could cause bad data recovery. The results for this instance are as below.
> |      | GT label |  DLG  |  Ours  | DLG+Ours |
> |:----:|:--------:|:-----:|:-----:|:--------:|
> | PSNR |  19.12   | 8.05  | 9.13 |   10.73   |
> | SSIM |  0.780   | 0.030 | 0.007 |  0.106   |
>
> For the second setting, we could simply try different label types iteratively. If the labels could only be label smoothing or mixup, then we could directly try mixup type first. We tested it on Resnet18 for 200 images with label smoothing augmentation, getting 100% accuracy. Image reconstruction results have no obvious difference from Table 3.
>
> In summary, the mathematical analysis has demonstrated that **knowing nothing about the label is not enough to recover smoothed labels**, it is not the limitation of our algorithm, but the limitation of the information contained in gradients. Inaccurate labels could cause bad image reconstruction in general, and it is also related to the extent to which the label diverges from the ground truth.

---

> > ### Comment · Reviewer_wnMY · 2023-11-20
> > **Thank You**
> >
> > Thanks for the rebuttal. It well answers my questions. I would like to moderately raise the score.

---

> > > ### Author Response · Authors · 2023-11-20
> > > **Thanks for your support**
> > >
> > > Dear reviewer wnMY,
> > >
> > > We are happy that our rebuttal provides more information about the new settings, which settles the concern. Your highly valuable comments benefit us a lot.
> > >
> > > Best,
> > >
> > > Authors of submission 5392.

---

### Official Review · Reviewer_xaET · 2023-10-29

**Soundness:** 3 good
**Presentation:** 2 fair
**Contribution:** 3 good
**Rating:** 6
**Confidence:** 2

**Summary:**

This paper introduces a framework designed to address the label recovery problem while incorporating features such as label smoothing and mixup. Their experimental results demonstrate the adaptability of their gradient inversion attacks in practical, real-world scenarios.

**Strengths:**

Seems that the performance is good compared with previous methods.

**Weaknesses:**

The introductory background and algorithm derivation may lack clarity, potentially causing individuals unfamiliar with this field to become confused.

**Questions:**

Why the label vector can be viewed as a function of the gradient in equation (3)?

---

> ### Author Response · Authors · 2023-11-14
>
> Thanks for the time you spent on our manuscript!
>
> 1. *“Introductory background may lack clarity.”* Due to the 9-page limitation, we eliminated some contents in the introduction and related work, which sacrifices some clarity. The main contribution of this work is to propose a novel algorithm to recover soft labels from gradients analytically, especially when the fully connected layer does not have the bias term, while previous works could only handle one-hot labels.
> 2. *“Algorithm derivation may lack clarity. Why the label vector can be viewed as a function of the gradient in equation (3)?”* The problem is to calculate the ground-truth label, and we only know two conditions: **the gradients** of the last layer $\mathbf{g}$, and the ground-truth input is **a scaled vector** of the gradients. If we know the ground-truth scalar $\lambda_r^\ast=p_r-y_r$, then we know the ground-truth inputs $\mathbf{x}^\ast$. Put it in the last layer, we get the post-softmax probability $\mathbf{p}$, and the ground-truth label could be recovered as the first line in Eqn. (3). Just because we do not have access to the exact value of $\lambda_r^\ast$, we set it as $\lambda_r$. Different $\lambda_r$ will generate different input $\mathbf{x}$, therefore different $\hat{p_i}$, and finally different $\hat{y_i}$. That is why $y_i$ is a function of $\lambda_r$, as shown in Eqn.(3),(4). Hope this could be helpful.
>
>
> Please do not hesitate to post other questions about the details if you still have any. We are happy to explain more.

---

> ### Author Response · Authors · 2023-11-21
> **Hope to get the feedback**
>
> Dear reviewer xaET,
>
> Sorry for bothering you. We genuinely appreciate the time and effort you've invested in our paper. As our rebuttal has been submitted for a while, we do want to know whether our explanations have settled the concerns. We are eager to have further discussions, so please let us know if you have additional feedback.
>
> Best,
>
> Authors of submission 5392.

---

> > ### Comment · Reviewer_xaET · 2023-11-22
> >
> > Thanks for the authors' response. It answers my questions here and I would like to raise my score.

---

> > > ### Author Response · Authors · 2023-11-23
> > > **Thanks for your support**
> > >
> > > Dear reviewer xaET,
> > >
> > > We are delighted to know that our explanation helps.  We do appreciate the support you provide for our work.
> > >
> > > Best,
> > >
> > > Authors of submission 5392.

---

### Official Review · Reviewer_3Kd5 · 2023-10-30

**Soundness:** 3 good
**Presentation:** 3 good
**Contribution:** 3 good
**Rating:** 8
**Confidence:** 4

**Summary:**

This paper introduces a novel algorithm to reconstruct training data in a more realistic scenario where augmented soft labels are utilized during training. Specifically, this paper focuses on the recovery of ground-truth augmented label and last-layer features in gradient inversion attacks instead of those with hard label constraints. Through the analysis of the gradients of cross-entropy loss and introduced variance loss, the proposed algorithm can tackle the recovery of soft labels. Extensive evaluations on various datasets demonstrate the effectiveness of the proposed algorithm.

**Strengths:**

1. The paper is well-organized and easy to follow.
2. The problem of soft label recovery can be challenging and interesting.
3. The proposed variance loss seems simple and natural.
4. The image reconstruction results of FCN regardless of the bias term seem promising.

**Weaknesses:**

1. The reason why the proposed variance loss can lead to the global minimum is not well-explained.
2. The authors should include more comparison with other baselines besides iDLG in label recovery evaluations, such as Table 1.
3. Lack details of the metrics to measure the correctness of soft label recovery.
4. The designing of searching procedure needs more explantation, such as the starting point of $\pm 1$.

**Questions:**

1. Please provide more discussion of variance loss.
2. Please provide more comparisons in label recovery evaluation.
3. Please explain the metrics in soft label recovery.
4. Please provide more discussion of the searching procedure in Section 3.3.

---

> ### Author Response · Authors · 2023-11-14
>
> Thanks for your detailed review!
>
> 1. *“Please provide more discussion of variance loss.”*  It is reasonable to doubt the strength of the simple variance loss. Here, we provide an intuitive explanation. After successfully simplifying the vector-recovery problem into one-value picking, **the label distribution is constrained and can only change in the dimension of $\lambda_r$.** Let's say we pick $\mathbf{g}_0$ as the initialization and have found the ground truth $\lambda_r^\ast$. Under real circumstances, according to the property of the SoftMax function, if we move $\lambda_r$ away from  $\lambda_r^\ast$, each entry will change at a different speed or in a different direction, and they will only meet once, as shown in Fig. 2 of the paper. For Mixup, the result of such a movement is similar. Therefore, for multi-class classification tasks, it is nearly impossible for all-but-the-top entries to firstly meet at $\lambda_r^\ast$, change with different speed and direction, and eventually meet again under another $\lambda_r$. Extensive experiments also demonstrate the performance.
> 2. *“Please provide more comparisons in label recovery evaluation.”* As stated in the paper, previous works mostly focus on recovering batched one-hot labels, and they are all unable to handle soft labels. The comparison with iDLG in Table 1 aims to show that our algorithm, designed for soft labels though, could still handle one-hot labels with identical performance as IDLG. **Currently we do not have other methods handling soft labels as baseline.**
> 3. *"Please explain the metrics in soft label recovery."* If we execute the one-hot label recovery, it is easy to define the accuracy as we could simply report the index of the class to check whether it is the case. For soft labels, the values are continuous, therefore we define $\mathcal{L}_r$ to describe how close two vectors are.  Assuming $\mathbf{a}$, $\mathbf{b}$ are two vectors with $n$ entry, $\mathcal{L}_r$ refers to the sum of $L_1$ loss of every entry:
>
> $\mathcal{L}_r=\sum_{i=1}^{n}{\lvert\mathbf{a}_{i}-\mathbf{b}_{i}\rvert}$
>
> 4. *"The starting point of ±1 needs more explanation."* It is really a precious point. when searching, we understand that $\lambda_r^\ast=1/(p_r-y_r)$. Therefore, it is larger than $1$ or smaller than $-1$. The searching starters correspond to the range of value. In codes, to increase efficiency and accuracy, we always pick the entry $i$ with the largest gradients (abstract value). From experiments, with such picking the ground-truth $\lambda_r^\ast$ is mainly in the range of [-10,10]. Detailed algorithms are also attached in Section E.

---

> > ### Comment · Reviewer_3Kd5 · 2023-11-20
> > **Thank you for your response**
> >
> > I would like to thank the authors for the rebuttal. My concerns have been addressed.

---

> > > ### Author Response · Authors · 2023-11-20
> > > **Thanks for raising the score**
> > >
> > > Dear reviewer 3Kd5,
> > >
> > > We are delighted to know that the concerns are fully addressed. We do appreciate the time you spent on our manuscript, as well as the support for our work.
> > >
> > > Best,
> > >
> > > Authors of submission 5392.

---

### Official Review · Reviewer_pnnF · 2023-11-04

**Soundness:** 3 good
**Presentation:** 3 good
**Contribution:** 2 fair
**Rating:** 6
**Confidence:** 2

**Summary:**

paper describes a simple method on label inference and also input reconstruction via analysis on one layer. this can be extended recursively to multiple layers.

**Strengths:**

the method is simple.

Eq.2 is a key equation. It can be checked with some algebra to be correct. from this equation, other things follow.

**Weaknesses:**

Eq.(3), there is a term (g_i/x^*), since g_i is a vector and x* is a vector this makes no sense.

the same goes for the term g_i/g_r which is 'vector divide by vector'.

Eq.(5) the authors said "top two entries" but does not precisely define what it means. could it be the two items in y_i with highest values?

table 1, table 2, table 3, table 4 and other results, error bars will be needed. otherwise it makes no sense to say one number has higher values than another.  repeated experiments e.g. using differently trained networks.

section 4.3 line 5, there is a typo.

**Questions:**

I am not very sure if this work would be of practical value if all the gradients and parameters needed to be known. for white box attack, parameters are known. however gradients should be known for this work to be valid. in practise, do we usually know gradients for each instance?

---

> ### Author Response · Authors · 2023-11-14
>
> Thanks for the detailed review.
>
> 1. For questions **"vector divide by vector"**, thanks for pointing it out. From Eqn. (2) we know the gradient $\mathbf{g}_i$ is a scaled input vector, so here the division is to calculate this scalar. It could be executed at item level, since they all get the same results. In codes, considering numerical stability, we use the mean value of item-level division. It is a precious point, and we will add footnotes in our manuscript to make it more clear.
> 2. For **“top two entries”**, your understanding is correct. The two items in $y_i$ with the highest values in our algorithm are regarded as entries involved in mixup, so we exclude them and penalize the variance of other entries.
> 3. For **error bars**, we added the standard deviation and **attached the full version of Table 3 and Table 4 in Section G** to make the comparisons more persuasive. For Table 2, we repeated the experiment to recover the smoothed label on untrained Lenet 5 times, 400 samples each time, and **the Acc. is 0.998 $\pm$ 0.004**. The first two tables aim to show the accuracy of our label recovery algorithm, where we tested 1000 samples with the specific label distribution on every network to demonstrate the performance. From experiments and our experience, it is quite stable.
> 4. For **practical value**, our work is based on federated learning framework, where in each step, **the central server collects uploaded gradients from several clients**, figures out the optimal updating direction, and then sends all updated (trained) parameters back to each client (this is the most common horizontal FL framework) [1][2]. As stated in threat model, the server could get access to the uploaded gradients. In practice, instance-wise gradients and batched gradients from clients are all possible. Our work is the first to handle soft labels, and we will extend it to batched situations in future works.
> 5. We do appreciate that you mentioned the typo. We will correct it in the latest version.
>
> [1] *Qiang Yang, Yang Liu, Tianjian Chen, and Yongxin Tong. Federated machine learning: Concept and applications. ACM Transactions on Intelligent Systems and Technology (TIST), 10(2):1–19, 2019*
>
> [2]*Ligeng Zhu, Zhijian Liu, and Song Han. Deep leakage from gradients. In Advances in neural information processing systems, volume 32, 2019.*

---

> ### Author Response · Authors · 2023-11-21
> **Hope to get the feedback**
>
> Dear reviewer pnnF,
>
> Sorry for bothering you. We appreciate the review you provided for our paper, and we have added the revised tables in the appendix. We are eager to have further discussions, please feel free to let us know if you have additional feedback!
>
> Best,
>
> Authors of submission 5392.

---

> > ### Comment · Reviewer_pnnF · 2023-11-22
> > **keep score**
> >
> > I like to thank the authors for putting in efforts to answer my queries and to edit their paper. I like to keep my scores.

---

> > > ### Author Response · Authors · 2023-11-22
> > > **Thanks for replying**
> > >
> > > Dear reviewer pnnF,
> > >
> > > We are delighted to receive the reply. Thanks again for your positive attitude towards our work.
> > >
> > > Best,
> > >
> > > Authors of submission 5392.

---

### Meta-Review · Area_Chair_L6h1 · 2023-12-08

**Metareview:**

The submission focuses on gradient inversion attacks when the label is soft, or constructed with augmentation schemes such as mixup that do not preserve integral labels.  The reviewers were unanimous in their opinion that the submission should be published at ICLR.  The simplicity of the variance loss was appreciated by the reviewers, as was the availability of code.  The reviewers also appreciated the quality of the reconstructions achieved by the method.  The rebuttal was handled well and resulted in several reviewers raising their scores.  A main point of improvement is that the description of why the variance loss works is still rather informal, e.g. in the response to reviewer 3Kd5.  The motivation in Section 3.2 is quite informal as well, and it feels like a proper theorem box is missing demonstrating that the loss gives a guarantee and tying it to the derivations in the previous section.

**Justification For Why Not Higher Score:**

The submission describes an extension of gradient inversion attacks to a slightly different label setting.  Such attacks are now featured in several main conferences, while this work is continued work in this direction.  Motivation for the proposed loss should be justified more formally in a theorem to move beyond an empirical contribution in order to better advance further research in this direction.

**Justification For Why Not Lower Score:**

unanimous recommendation for accept.

---

### Decision · Program_Chairs · 2024-01-16

Accept (poster)